# Confocal Microscopy Investigations of Biopolymeric PLGA Nanoparticle Uptake in *Arabidopsis thaliana* L. Cultured Cells and Plantlet Roots

**DOI:** 10.3390/plants12132397

**Published:** 2023-06-21

**Authors:** Giulia De Angelis, Camilla Badiali, Laura Chronopoulou, Cleofe Palocci, Gabriella Pasqua

**Affiliations:** 1Department of Environmental Biology, Sapienza University of Rome, P. le Aldo Moro 5, 00185 Rome, Italy; 2Department of Chemistry, Sapienza University of Rome, P. le Aldo Moro 5, 00185 Rome, Italy; 3Research Center for Applied Sciences to the Safeguard of Environment and Cultural Heritage (CIABC), Sapienza University of Rome, P. le Aldo Moro 5, 00185 Rome, Italy

**Keywords:** endocytosis, dynamin-independent endocytosis, PLGA nanoparticles, dynasore, cell cultures, root plantlets, *Arabidopsis thaliana*

## Abstract

To date, most endocytosis studies in plant cells have focused on clathrin-dependent endocytosis, while limited evidence is available on clathrin-independent pathways. Since dynamin a is a key protein both in clathrin-mediated endocytosis and in clathrin-independent endocytic processes, this study investigated its role in the uptake of poly-(lactic-co-glycolic) acid (PLGA) nanoparticles (NPs). The experiments were performed on cultured cells and roots of *Arabidopsis thaliana*. Dynasore was used to inhibit the activity of dynamin-like proteins to investigate whether PLGA NPs enter plant cells through a dynamin-like-dependent or dynamin-like-independent endocytic pathway. Observations were performed by confocal microscopy using a fluorescent probe, coumarin 6, loaded in PLGA NPs. The results showed that both cells and roots of *A. thaliana* rapidly take up PLGA NPs. Dynasore was administered at different concentrations and exposure times in order to identify the effective ones for inhibitory activity. Treatments with dynasore did not prevent the NPs uptake, as revealed by the presence of fluorescence emission detected in the cytoplasm. At the highest concentration and the longest exposure time to dynasore, the fluorescence of NPs was not visible due to cell death. Thus, the results suggest that, because the NPs’ uptake is unaffected by dynasore exposure, NPs can enter cells and roots by following a dynamin-like-independent endocytic pathway.

## 1. Introduction

The challenge for the next century is to find innovative solutions, both economically and environmentally sustainable ones, that can be applied to agriculture to face the limited natural resources (land, water, soil, etc.) and the increasing world population. The use of nanomaterials in agriculture would lead to optimizing fertilization by reducing nutrient losses, reducing the amount of chemicals dispersed into the environment, and increasing yield through pest management. In the last decades, the study of biopolymeric nanoparticles (NPs) has yielded promising results on their use in the agronomic field for the delivery of bioactive substances [1]. Biodegradable poly-(lactic-co-glycolic) acid (PLGA) NPs have shown great potential as drug delivery carriers in the fields of plant and environmental sciences as well as the medical field, which is attributed to their controlled release of bioactive compounds [2,3,4,5,6] and proven biocompatibility and biodegradability [7]. Nevertheless, little information is available on the absorption, uptake, translocation, and interaction processes of NPs in plant systems [8]. Understanding these mechanisms would provide a crucial basis for assessing the effects of nanoparticles on agriculture, food safety, and the environment, but racking these vehiculation systems inside plants does require complex probe tagging strategies [9]. Most of the studies on PLGA NPs have been carried out in mammalian and human systems, and little information is available about the PLGA NPs’ uptake in plant cells [10,11,12]. Transmission electron microscopy (TEM) analyses carried out on grapevine cells suggest that PLGA NPs are internalized by endocytic vesicles [10].

The most investigated and validated mechanism for cargo molecules internalization in cells seems to be clathrin-mediated endocytosis (CME); however, the endocytic pathway in plants is poorly characterized and its hypothesized mechanism is largely inferred from studies in mammalian and yeast systems, where CME components are highly conserved [13]. CME begins with the formation of clathrin-coated membrane invaginations, also known as clathrin-coated pits. An essential protein for vesicle detachment is dynamin, a GTPase protein [14]. Nonetheless, in addition to the clathrin-dependent endocytosis pathway, different studies have revealed several clathrin-independent pathways in plant cells, which are also not all dependent on dynamin [15]. Furthermore, clathrin-independent pathways have been implicated in studies on tobacco (*Nicotiana tabacum*) cells grown in suspension cultures [16] and in epidermal cells of *Arabidopsis* root [17] but neither of these pathways has been well characterized with respect to the identity of the handled cargo or the endocytic mechanisms below the epidermis. Through fluorescence microscopy, it has been established that PLGA NPs can enter grapevine leaf tissues through stomata openings and that they can be absorbed by the roots and transported to the shoots through vascular tissues. In addition, in grapevine cells, PLGA NP-containing vesicles were observed exclusively in the cytoplasm. The absence of PLGA NPs in the vacuoles suggested that clathrin-independent endocytosis may be the main internalization pathway [10]. The results obtained by administering PLGA NPs to grapevine cells, after treatment with the clathrin-dependent endocytosis inhibitors ikarugamycin and wortmannin, led to the hypothesis that PLGA NPs uptake largely follows the clathrin-independent endocytic route [11]. However, the question remained open because these inhibitors are not chemical modulators of protein trafficking components involved in the early steps of CME [18]. Ikarugamycin is an antibiotic with antiprotozoal activity that exhibits strong cytotoxicity [19]. It causes the redistribution of the clathrin heavy chain and the adaptor protein 2 [20]; however, its precise mechanism of action remains unclear. Wortmannin, on the other hand, is an inhibitor of phosphatidylinositol-3 kinase, an essential component for the formation of internal vesicles [21]. Treatment with wortmannin leads to a perturbation of receptor recycling in yeasts and mammals [22]. Further studies on the endocytic mechanisms in plant cells and tissues are indispensable to shed light on the uptake of NPs in plants.

It is important to highlight that not all the clathrin-independent pathways require dynamin activity. On the contrary, the clathrin-dependent pathway is closely linked to the activity of dynamin [14]. In 2006, Macia and co-workers screened about 16,000 small molecules identifying dynasore, which interferes in vitro with the GTPase activity of dynamin1, dynamin2, and Drp1, the mitochondrial dynamin, but not with other small GTPases. Dynasore acts as a potent inhibitor of the endocytic pathways known to depend on dynamin by blocking the GTPase activity of dynamin, thus preventing vesicles from detaching from the membrane and rapidly blocking coated vesicle formation in the transferrin uptake of HeLa cells [23]. In the present study, the PLGA NPs’ uptake mechanism has been investigated in *A. thaliana* cultured cells and in 10-day-old plantlets. The PLGA NPs were loaded with the fluorescent probe coumarin 6 (Cu6-PLGA NPs) and administered in aqueous suspensions to cells and *A. thaliana* seedling roots. To clarify whether Cu6-PLGA NPs penetrated plant cells through the dynamin-like-dependent or dynamin-like-independent endocytic pathway, the inhibitor dynasore was used both in experiments on cultured cells and on plant roots.

## 2. Results

The endocytosis mechanisms of the PLGA NPs have been investigated by confocal microscopy on *A. thaliana* cell cultures. On day 10 of the subculture, in the exponential growth phase, the cultured cells appeared spherical in shape and were collected for the uptake experiment (Figure 1). *A. thaliana* cell cultures were incubated for 10 min with Cu6-PLGA NPs (30 nm) at a concentration of 15 mg L^−1^. We observed the presence of small highly fluorescent round bodies (endosomes) and a low diffuse background within the cells (Figure 1). With increasing incubation time, the endosomes appeared to coalesce but their diameter increased after 30 min. Furthermore, the PLGA NPs were accumulated in the cytoplasm and seemed to be stable over time, since 24 h after the Cu6-PLGA NPs treatment, the fluorescence distribution pattern was similar to the one observed after 60 min.

To clarify whether Cu6-PLGA NPs penetrated plant cells through the dynamin-like-dependent or dynamin-like-independent endocytic pathway, the inhibitor dynasore was used. Cell treatments with 80 µM or 160 µM dynasore for 10 min did not prevent NP uptake, as revealed by the presence of fluorescence emission in the cytoplasm and in spherical vesicles (Figure 1g,h).

The investigation of PLGA NPs’ endocytosis mechanism was also carried out in the *A. thaliana* plantlets. After 10 min of treatment with 15 mg L^−1^ Cu6-PLGA NPs, fluorescence was visible in the hair roots (Figure 2a). After 30 and 80 min, fluorescence was widespread through almost all cells of the epidermis (Figure 2b,c). Cu6-PLGA NPs were additionally observed at the level of the epidermis after 5 h (Figure 2d). Plantlet roots were treated with 80 or 160 µM dynasore for 30 and 60 min before the addition of 15 mg L^−1^ Cu6-PLGA NPs. Nevertheless, the NPs’ uptake was not prevented by dynasore treatment, as revealed by fluorescence in the epidermis (Figure 3a,b,d,e). Even when treating the roots with a higher concentration of dynasore (320 µM) for 30 and 60 min before the Cu6-PLGA NPs treatment, intense fluorescence continued to be visible in the root epidermis (Figure 3c,f). On the other hand, the fluorescence signal was absent when the plantlet roots were treated with 320 µM dynasore for 120 min. In this case, no Cu6-PLGA NPs were observed in the root cells (Figure 4c). The cytotoxicity test with propidium iodide was performed on root cells to evaluate the inhibitor cytotoxicity. After treatment with dynasore at the highest concentration (320 µM) for 120 min, the root cells were not viable (Figure 4d). Consequently, the absence of fluorescence in the epidermis root cells was not caused by the inhibition of Cu6-PLGA NPs’ uptake. Instead, Cu6-PLGA NPs were not able to enter cells because the latter were dead. It is known that propidium iodide is a membrane-impermeable dye that is generally excluded from viable cells and, conversely, fills the cytoplasm with non-viable cells [24]. Interestingly, the cytotoxicity test performed on roots treated with 80 µM dynasore for 120 min showed that the roots were viable (Figure 4b) and, after Cu6-PLGA NPs treatment, fluorescence was clearly visible in the epidermis (Figure 4a). These results provide evidence that dynasore did not inhibit the uptake of Cu6-PLGA NPs, suggesting that the internalization of NPs follows the dynamin-like-independent pathway since the GTPase activity of dynamin, which causes the detachment of neo-formed endocytic vesicles, was not required.

## 3. Discussion

To date, little information on PLGA NP internalization mechanisms in plant cells is reported in the literature. Previous studies carried out on *Vitis vinifera* highlighted that PLGA NPs may enter suspended cells as well as leaf tissues through stomata openings (in addition to being absorbed by the roots) as revealed by fluorescence and transmission electron microscopy (TEM) [10]. In addition, TEM observations by Palocci and co-workers [11] showed the formation of endocytic vesicles containing PLGANPs which, undergoing coalescence, form larger vesicles located in the cytoplasm. The uptake of NPs in plants is species-specific and depends on the NPs’ properties, such as size, porosity, hydrophobicity, and surface area. For example, it is known that Au NPs are taken up by tobacco but not by wheat [20,21,22,23,24,25]. The porous nature of cell walls allows NPs to cross them by binding to protein carriers via aquaporins, ion channels, endocytosis, or by piercing the cell membrane and creating new pores [26,27,28]. Palocci and co-workers [11] suggested a role of the cell wall in the size selection of PLGA NPs, showing that PLGA NPs with a diameter smaller than 50 nm were able to penetrate *V. vinifera* cells. In *Arabidopsis* cells, PLGA NPs of 30 nm can enter cells. In suspension cultures of *Chenopodium album* L., *Dioscorea deltoidea* Wall., and *Medicago sativa* L., the mean size limit was found to vary between 2.4 and 3.8 nm [29]. Bandmann and Homann [30] proved that the cell wall blocks the uptake of nanobeads larger than 100 nm in BY-2 tobacco cells. The reason for these differences between the various species could depend on the structure, organization, and interactions of cellulose, hemicelluloses, pectins, structural proteins, and lignin of the cell wall [31]. Particularly, in the primary wall, pectins are known to regulate porosity through the galactan and arabinan side chains of rhamnogalacturonan [32]. TEM analyses conducted on *M. sativa* suspension cultures showed that PLGA NPs passed through the cell wall and accumulated into the cytoplasm and nucleus [33]. After crossing the cell wall, NPs then cross the membrane by endocytosis [34], a process involving the internalization of extracellular materials or plasma membrane proteins into the cell through a series of vesicular compartments [35]. Similar to animal cells, the major route for cell uptake in plants is clathrin-mediated endocytosis (CME), which starts with the invagination of the clathrin-coated membrane [36]. Subsequently, dynamin assembles as a collar on the neck of a budding pit, and the conformational change accompanying GTP hydrolysis causes a constriction and scission of the neck [37]. However, several studies have revealed alternative clathrin-independent pathways, including membrane microdomain-associated endocytosis, fluid-phase endocytosis, and phagocytosis-like uptake of rhizobia in plants [14,38]. For example, in *Arabidopsis*, the membrane microdomain-associated flotillin1 (Flot1) is involved in clathrin-independent endocytosis, as demonstrated by studying a transgenic green fluorescent protein—flotillin1—in *A. thaliana* plants with confocal microscopy analysis and immunogold labeling transmission electron microscopy [17]. It was also discovered that some clathrin-independent endocytosis pathways are dynamin-independent [39] and that some members of the ADP-ribosylation factor (Arf) and Rho subfamilies of small GTPases have key roles in regulating different pathways of clathrin-independent endocytosis [40].

Moreover, Macia and co-workers [23] showed that dynasore acts as a potent inhibitor of dynamin-related endocytic pathways in animal cells. The supplementation of 80 µM dynasore inhibited the uptake of transferrin within 1–2 min of treatment in HeLa and astrocytes cells, presumably because it was limited by the diffusion of the molecule to the coated pits on which dynasore acted. In fact, electron microscopy images of cells treated for 10 min with 80 µM dynasore showed a large number of coated pits at partial stages of late assembly that remained linked to the plasma membrane by either narrow or wide necks. They also demonstrated that extended incubation (48 h) of cells with 80 µM dynasore was not toxic and reversible. Up to now, however, there are no studies demonstrating the effect of dynasore as an inhibitor of NPs’ endocytosis in plants.

In our study, Cu6-PLGA NPs were taken up by roots of *A. thaliana*. Fluorescence was observed in the root hairs and the epidermis after 10 min; between 30 and 80 min after treatment, fluorescence was visible as a diffuse signal in the root hairs and in all cells of the epidermis. After 5 h, fluorescence appeared less intense, and although NPs were not observed in the vascular cylinder, we speculate that they translocated into the stem. 

NPs are known to be able to translocate into the plant and they can also move to organs that are quite far from the site of uptake, although the related mechanisms remain to be elucidated. The choice of size and chemical characteristics of the NPs determines the delivery method and target organ in which uptake occurs most efficiently. The ability to translocate within the plant becomes an important factor to consider to achieve the required biological effects in the tissue or organ of interest. It has been demonstrated that the translocation route is mainly determined by the size, surface charge, and chemical properties of NPs [41]. NP uptake can occur through the pores of the root epidermal cell walls (size around 5–30 nm), the so-called apoplastic route [42]. NPs that enter the root may be subjected to capillary forces and osmotic pressure, and diffuse through the apoplast to reach the endodermis [42,43]. The symplastic pathway is another route through which NPs are taken up by plants, moving via the inner side of the plasma membrane. Once inside the cell, NPs can enter adjacent cells through plasmodesmata (20–50 nm diameter channels) [44] and/or exocytosis [44,45]. NPs translocate in the plant through the xylem and phloem. The phloem forms a porous arrangement with diameters ranging from 200 nm to 1.5 μm in most plants [46,47,48]. Metallic and polymeric NPs have been shown to be efficiently translocated through the phloem [49,50,51,52,53]. Most NPs are believed to be able to pass through the phloem unless they are aggregated. The xylem pore sizes range from 43 to 340 nm [54,55,56]. Thus, NP translocation may be limited by pore size and NPs that potentially accumulate in cells [57,58]. It has been shown that negatively charged NPs can be transported from roots to shoots through the xylem [59,60]. Valletta and co-workers demonstrated that *V. vinifera* plant roots treated with Cu6-PLGA NPs accumulated NPs in the shoot tissues after 48 h of treatment [10]. Moreover, confocal microscopy images of the sugarcane roots showed fluorescent signals of zein NPs along the epidermal layer, and the translocation of these NPs followed from the cortex to the endodermis and then to the leaves [9].

For the first time in this study, the treatment of *A. thaliana* cell suspension cultures and plantlet roots with dynasore was tested. The treatment with 80 µM and 160 µM dynasore for 10 min did not prevent the uptake of Cu6-PLGA NPs. The treatment of *A. thaliana* roots with 80 µM or 160 µM dynasore for 30, 60, and 120 min or with 320 µM dynasore for 30 and 60 min did not prevent the uptake of Cu6-PLGA NPs and the root cells were found to be viable. On the contrary, cells treated with 320 µM dynasore for 120 min were not viable.

Overall, the results of this study suggest that Cu6-PLGA NPs are uptaken by *A. thaliana* cell suspension cultures and plantlet roots through a dynamin-like-independent pathway.

## 4. Materials and Methods

### 4.1. PLGA NPs Preparation

As previously reported [61], PLGA NPs with a diameter of 30 nm, empty, or entrapping coumarin 6, were prepared by nanoprecipitation using a flow-focusing capillary reactor made of stainless steel capillary tubes with an internal diameter of 256 μm, shown in Figure 5. In a typical preparation, 2.5 mL of a 2 mg mL^−1^ PLGA solution in acetone were used. The organic phase and water phase flow rate were fixed at 50 and 2000 µL min^−1^, respectively. Additionally, the fluorescent NPs were prepared by using coumarin 6 with a ratio of 1:50 *w*/*w* with respect to PLGA. The organic and water phase meet in a cross junction, and nanoprecipitation occurs in the successive outlet mixing channel, from the end of which PLGA-based NPs can be recovered. In total, 50 mL of suspension were collected. The organic solvent (acetone) was eliminated under reduced pressure, and PLGA-based NPs were stored at 4 °C until use.

### 4.2. Plant Growth Conditions

*A. thaliana* (L.) Heynh var. Columbia (Col-0) seeds were used for all experiments. The plantlets were vertically grown in Petri dishes (90 mm diameter) and in aseptic conditions for 10 days after sowing on solid Murashige and Skoog medium in a controlled environment chamber under a photoperiod of 16/8 h (light/dark) and at 26 ± 1 °C (photon flux density of 70 µmol m^−2^ s^−1^).

### 4.3. Cell Suspension Cultures

According to May and Leaver [62], callus formation was induced by placing stem explants on agarized medium (Gamborg B5, Glc 2% [*w*/*v*], agar 0.8% [*w*/*v*], Mes 0.5 g L^−1^, 2,4-D 0.5 mg L^−1^, kinetin 0.05 mg L^−1^), which was maintained in a growth chamber at 26 ± 1 °C under continuous darkness and subcultured every 25 days. Suspension cultures were obtained by inoculation of 300 mg of rapidly dividing, friable, white callus into 50 mL of the Murashige and Skoog medium containing 0.5 mg L^−1^ naphthaleneacetic acid, 0.05 mg L^−1^ kinetin, 3% (*w*/*v*) sucrose, and incubated on a rotary shaker at 100 rpm [58]. Cell suspensions were subcultured every 25 days for 3 months by decanting 60% of the medium and replacing it with fresh medium. The pH of all media was adjusted to 5.6 by adding 1 N NaOH before autoclaving at a temperature of 121 °C and 1 atm for 20 min.

### 4.4. PLGA NPs Uptake Experiments in Cultured Cells

PLGA NPs in aqueous suspension were sonicated for 30 min before adding them to *A. thaliana* Col-0 cell suspensions. For cultured cell experiments, Cu6-PLGA NPs were added to the liquid culture media at a final concentration of 15 mg L^−1^ (at day 10 of subculture), incubating under continuous darkness on a rotary shaker at 100 rpm. The observations were carried out at different times after Cu6-PLGA NPs treatment: 0 min, 10 min, 30 min, 1 h, 2 h, and 24 h. In accordance with the work of Macia and co-workers [23], dynasore (purchased from Sigma-Aldrich, Milan, Italy) was added to cultured cells to a final concentration of 80 µM to investigate whether the Cu6-PLGA NPs’ endocytic process is prevented by this inhibitor. Since the literature reports no article on the use of dynasore in plant cells, a higher concentration of 160 µM was also tested. Cells were incubated with dynasore for 10 or 30 min, after which Cu6-PLGA NPs were added to cell suspensions (15 mg L^−1^ final concentration) and cells were observed by confocal microscopy after 10 min incubation time.

### 4.5. PLGA NPs Uptake Experiments in Plantlet Roots

PLGA NPs were prepared as described for cultured cell experiments. For root uptake experiments, *A. thaliana* plantlets (10 days after seedling) were transferred from an agarized medium to a multiwell plate (6 plates) filled with 5 mL of liquid culture media, being careful not to damage the roots, and Cu6-PLGA NPs were added to the medium at a final concentration of 15 mg L^−1^ under continuous darkness on a rotary shaker at 100 rpm. To evaluate the Cu6-PLGA NPs’ absorption, confocal microscopy observations were carried out at 0 min, 10 min, 30 min, 80 min, and 5 h. To investigate the root endocytic uptake of NPs, dynasore was added to plantlets at a final concentration of 80, 160, or 320 µM. Specifically, plantlets were incubated with dynasore for 30 min, 60 min, and 120 min, and then the roots were treated with 15 mg L^−1^ of NPs and observed by confocal microscopy 10 min later.

### 4.6. Cytotoxicity Test on Dynasore Treated Plantlets

The fluorescent dye propidium iodide is commonly used to stain plant roots [63,64]. A 12.5 μg mL^−1^ propidium iodide water solution was used in this case. Plantlet roots were treated with different dynasore concentrations (80 and 320 µM) and with Cu6-PLGA NPs 120 min after dynasore treatment. Then, they were mounted on microscope slides and root cell viability was assessed by confocal microscopy 10 min after the Cu6-PLGA NPs’ treatment. The cells were considered nonviable if propidium iodide filled the cytoplasm and viable if propidium iodide was excluded. The excitation wavelength of the reader was 530 nm with an emission filter of 620 nm.

### 4.7. Confocal Microscopy Analysis

In order to investigate the uptake of Cu6-PLGA NPs by cells and roots, differential interference contrast (DIC) and confocal microscopy were used. The analyses were performed on an inverted Z.1 microscope (Zeiss, Germany) equipped with a Zeiss LSM 700 spectral confocal laser scanning unit (Zeiss, Germany). For coumarin 6 detection, a 488 nm, 10 mW solid laser with an emission wavelength of 520 nm was employed, while for propidium iodide detection, a 555 nm, 10 mW solid laser with an emission wavelength split at 550 nm/LP 640 was employed.

## 5. Conclusions

In the present study, we evidenced that Cu6-PLGA NPs rapidly penetrated cultured cells and plantlet roots of *A. thaliana*. It was observed that dynasore did not inhibit the uptake of Cu6-PLGA NPs in cells and roots, suggesting that the NPs’ internalization may occur by a dynamin-like-independent pathway and, consequently, by a clathrin-independent pathway, in agreement with our previous studies performed on *V. vinifera*. The diverse uptake mechanisms of NPs could affect the rate of uptake and translocation within the plant as well as the potential sites of accumulation. Therefore, the results obtained in the present work add new, relevant information to advance studies in the field of PLGA NPs’ uptake. Further studies are essential to better understand the translocation of PLGA NPs through plant tissues and determine accumulation sites, especially with the aim of using PLGA NPs as carriers for bioactive molecules (i.e., elicitors, antifungals) for applications in the agronomic field. Moreover, the long-term effects on plants as well as soil will have to be carefully evaluated and quantified.

## Figures and Tables

**Figure 1 plants-12-02397-f001:**
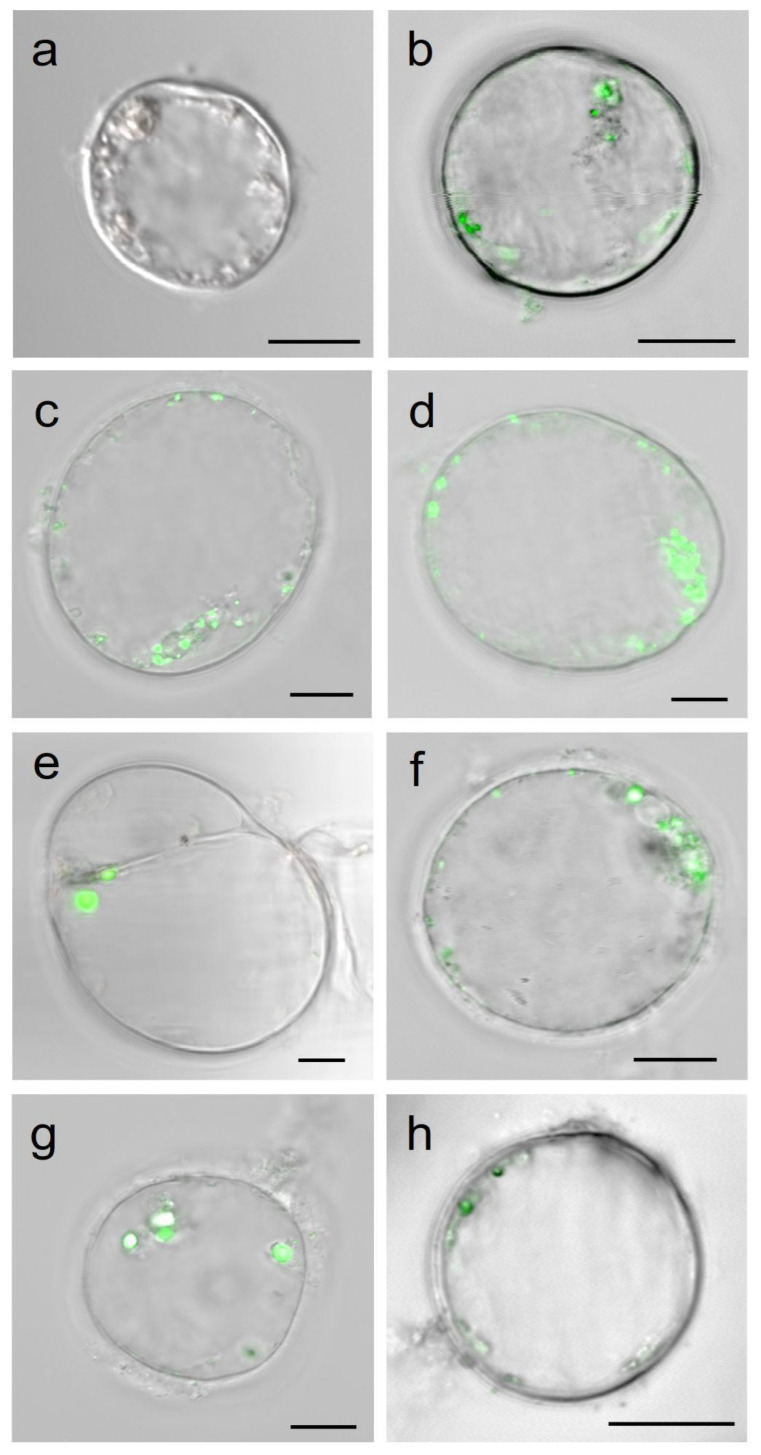
Cell suspension cultures of *A. thaliana* treated with 30 nm NPs and observed with a confocal microscope at different times ((**a**) 0 min; (**b**) 10 min; (**c**) 30 min; (**d**) 1 h; (**e**) 2 h; (**f**) 24 h). Fluorescence is localized within spherical bodies. (**g**,**h**) *A. thaliana* suspended cells observed by confocal microscopy after treatment with dynasore at 80 µM (**g**) or 160 µM (**h**) for 10 min before the addition of 15 mg L^−1^ NPs. Fluorescence in both cytoplasm and vesicles is still visible. Merged (bright field/fluorescence) images are shown. The scale bars represent 10 µm.

**Figure 2 plants-12-02397-f002:**
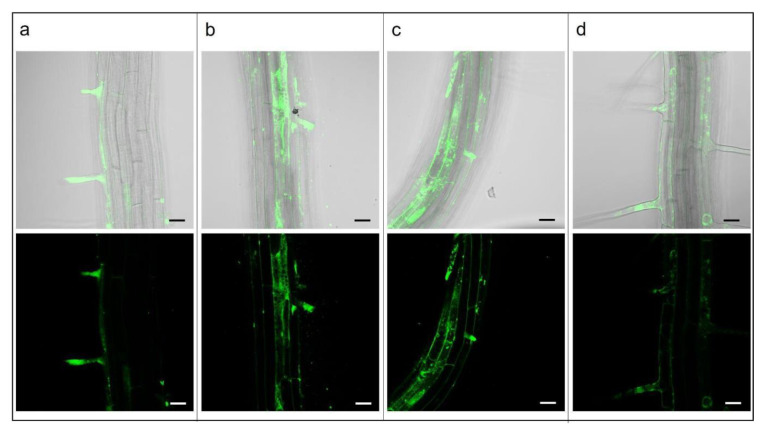
*A. thaliana* plantlet roots treated with 30 nm NPs (15 mg L^−1^) and observed by confocal microscopy at different times of Cu6-PLGA NPs treatment (10, 30, 80 min, and 5 h). Merged (bright field/fluorescence) and fluorescence images are shown. (**a**) *A. thaliana* root treated for 10 min with Cu6-PLGA NPs. Cu6-PLGA NPs penetrated through the root hairs; (**b**) *A. thaliana* root treated for 30 min with Cu6-PLGA NPs; (**c**) *A. thaliana* root treated for 80 min with Cu6-PLGA NPs. Cu6-PLGA NPs were not only visible in root hairs but in almost all cells of the epidermis. (**d**) *A. thaliana* roots treated for 5 h with Cu6-PLGA NPs. Cu6-PLGA NPs penetrated into the root epidermis through the root hairs and remained at the level of the epidermis. The scale bars represent 20 µm.

**Figure 3 plants-12-02397-f003:**
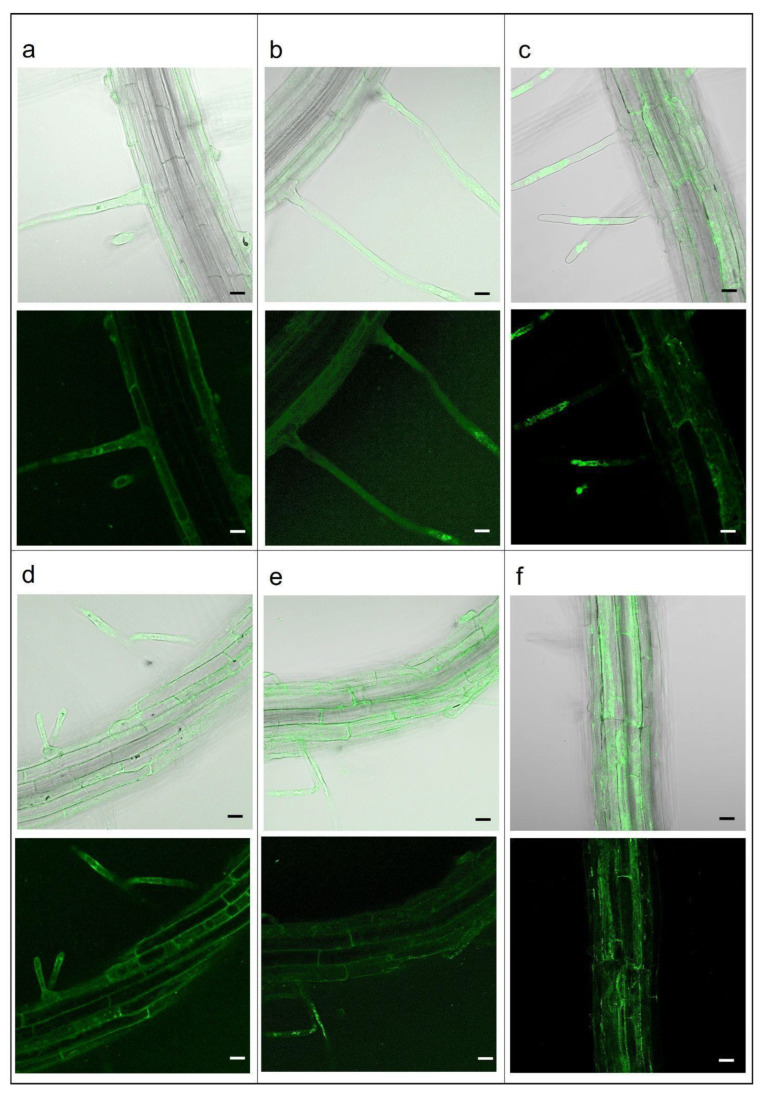
*A. thaliana* seedling roots treated with 80, 160, or 320 µM dynasore for 30 and 60 min before the addition of 15 mg L^−1^ Cu6-PLGA NPs for 10 min. Merged (bright field/fluorescence) and fluorescence images are shown. (**a**) Treatment with 80 µM dynasore for 30 min before Cu6-PLGA NPs treatment; (**b**) treatment with 160 µM dynasore for 30 min before Cu6-PLGA NPs treatment; (**c**) treatment with 320 µM dynasore for 30 min before Cu6-PLGA NPs treatment; (**d**) treatment with 80 µM dynasore for 60 min before Cu6-PLGA NPs treatment; (**e**) treatment with 160 µM dynasore for 60 min before Cu6-PLGA NPs treatment; (**f**) treatment with 320 µM dynasore for 60 min before Cu6-PLGA NPs treatment; Cu6-PLGA NPs penetrated the root epidermis through the root hairs and remained at the level of the epidermis. The scale bars represent 20 µm.

**Figure 4 plants-12-02397-f004:**
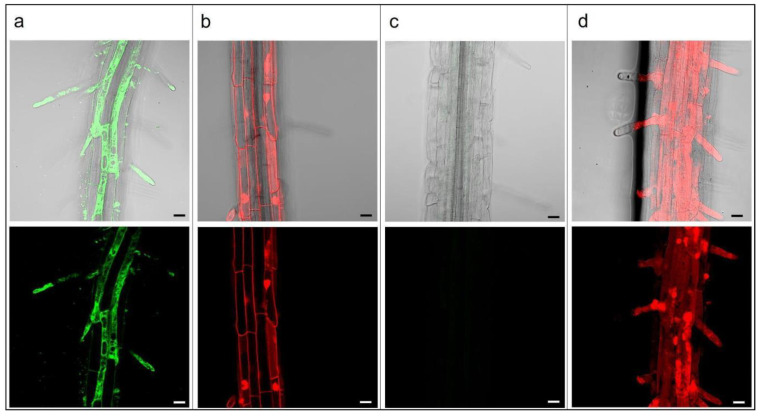
*A. thaliana* seedling roots treated with 80 and 320 µM dynasore for 120 min before propidium iodide treatment for cytotoxicity test. (**a**) Treatment with 80 µM dynasore for 120 min before Cu6-PLGA NPs treatment; (**b**) treatment with 80 µM dynasore for 120 min before propidium treatment; almost all cells are found to be viable; (**c**) treatment with 320 µM dynasore for 120 min before Cu6-PLGA NPs treatment; (**d**) treatment with 320 µM dynasore for 120 min before propidium treatment; almost all cells are found to be nonviable. Nonviable cells are red. Merged (bright field/fluorescence) and fluorescence images are shown. The scale bars represent 20 µm.

**Figure 5 plants-12-02397-f005:**
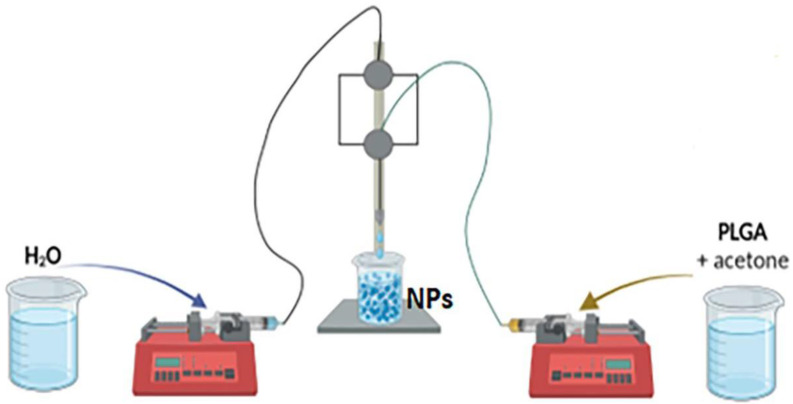
Schematic representation of the microfluidic reactor used for the synthesis of PLGA NPs.

## Data Availability

Not applicable.

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
