# Peer review of "Confocal Microscopy Investigations of Biopolymeric PLGA Nanoparticle Uptake in Arabidopsis thaliana L. Cultured Cells and Plantlet Roots"

_plants, 2023, doi:10.3390/plants12132397_

Round 1

Reviewer 1 Report

1.      In subsection 4.1, providing more information about the flow-focusing capillary reactor would be helpful, such as the dimensions and materials used. This information would be useful for readers who want to replicate the experiment.

2.      In subsection 4.2, it would be helpful to provide information about the size of the Petri dishes used for growing the plantlets, as this can affect plant growth.

3.      In subsection 4.4, it would be helpful to provide more information about the characteristics of the cultured cells, such as their growth rate and morphology. Additionally, it would be useful to provide information about the imaging equipment used for observing the cells.

4.      In subsection 4.5, it would be helpful to provide information about the age of the plantlets used for the experiments, as this can affect their ability to absorb nanoparticles. Additionally, it would be useful to provide information about the imaging equipment used for observing the plant roots.

5.      The study primarily focuses on PLGA NPs internalization in plant cells and does not address potential toxic effects or long-term impacts on plant growth and development.

6.      While the authors provide a comprehensive literature review on PLGA NPs uptake mechanisms in plants, the study does not contribute much novel information beyond confirming previous findings.

7.      The study uses V. vinifera, Arabidopsis, and tobacco as model plants, but it is unclear how generalizable the results are to other plant species.

8.      The methods used in the study are mainly focused on imaging and microscopy techniques, and there is a lack of quantitative data on the uptake efficiency of PLGA NPs.

9.      The authors do not provide much discussion on the potential applications or implications of their findings for nanotechnology or agriculture.

10.   The investigation was only carried out on one plant species (A.thaliana), which may not reflect other plant species' uptake mechanisms.

11.   The study only looked at the endocytosis mechanism of PLGA NPs, but did not investigate other potential pathways that may contribute to the uptake of NPs in plant cells.

12.   The study relied on fluorescence imaging to visualize the uptake of NPs, which may not provide a comprehensive understanding of the mechanisms involved in NP uptake.

13.   The study used dynasore as an inhibitor to investigate the involvement of dynamin in the uptake of NPs, but other pathways that are not dependent on dynamin activity may also be involved in NP uptake.

14.   The cytotoxicity test performed on root cells to evaluate the inhibitor cytotoxicity was limited to one dye (propidium iodide). Other methods of assessing cytotoxicity may be necessary to evaluate the safety of NP uptake in plant cells fully.

Author Response

1) In subsection 4.1, providing more information about the flow-focusing capillary reactor would be helpful, such as the dimensions and materials used. This information would be useful for readers who want to replicate the experiment.

Reply:  As suggested by the reviewer we have added more information on NPs preparation, also providing a description of the microfluidic apparatus employed for their preparation.

2) In subsection 4.2, it would be helpful to provide information about the size of the Petri dishes used for growing the plantlets, as this can affect plant growth.

Reply: As suggested by the reviewer the size of Petri dishes (diameter 9 mm) has been added. 

3) In subsection 4.4, it would be helpful to provide more information about the characteristics of the cultured cells, such as their growth rate and morphology.  Additionally, it would be useful to provide information about the imaging equipment used for observing the cells.

Reply: We agree with the reviewer, more information about cultured cell characteristics has been added to the results paragraph. The imaging equipment is described in subsection 4.7.

4) In subsection 4.5, it would be helpful to provide information about the age of the plantlets used for the experiments, as this can affect their ability to absorb nanoparticles. Additionally, it would be useful to provide information about the imaging equipment used for observing the plant roots.

Reply: Subsections 4.2 and 4.5 have been improved to clarify how the experiment was carried out. We used 10-day-old plantlets that have been transferred from an agarized medium to a multiwell plate where the experiment took place. The imaging equipment is described in subsection 4.7.

5) The study primarily focuses on PLGA NPs internalization in plant cells and does not address potential toxic effects or long-term impacts on plant growth and development.

Reply: We agree with the reviewer that the large scale use of NPs requires that potential toxic and long-term effects on plants, as well as soil, will have to be carefully evaluated and quantified. In this first phase of the experiments, however, our focus was on evaluating the uptake and toxicity occurring on plants in the early stages of development. We believe that such studies may provide a basis for future studies on adult plants.

6) While the authors provide a comprehensive literature review on PLGA NPs uptake mechanisms in plants, the study does not contribute much novel information beyond confirming previous findings.

Reply: The study of NPs internalization in plants is still at an early stage, therefore elucidating the mechanisms underlying the process will require a lot of research work. In plants, unlike animal cells, uptake mechanisms are related to the species involved and the structure of the cell wall. In this work, we have obtained results suggesting, for the first time, that the dynamin-independent pathway is the favored one for the internalization of PLGA NPs and that the dynamin-dependent pathways are not involved.

7) The study uses V. vinifera, Arabidopsis, and tobacco as model plants, but it is unclear how generalizable the results are to other plant species.

Reply: Analyses were carried out in previous works on V. vinifera and in this study on A. thaliana. V. vinifera cells are a system on which we have much experience as well as being a plant of huge agronomic interest, so the previous study focused on this species. When interesting results emerged that suggested that the internalization mechanism was clathrin-independent we decided to continue the survey on cultured cells and then the roots of the model plant A. thaliana.

A. thaliana as a model plant is a simplified system for conducting experiments in order to obtain insights regarding mechanisms that can be extended to other species with further specific analyses.

We do not exclude extending the analysis of uptake mechanisms to other plant species, as, unlike in animal cells, uptake mechanisms depend on the species and the specific structure of the cell wall.

8) The methods used in the study are mainly focused on imaging and microscopy techniques, and there is a lack of quantitative data on the uptake efficiency of PLGA NPs.

Reply: In this study, the objective was to verify whether or not NPs absorption occurred by measuring an all-or-nothing effect. We wanted to obtain results that would suggest whether or not dynamin was involved in the process, so the study was not aimed at quantifying the uptake efficiency of PLGA NPs.

9) The authors do not provide much discussion on the potential applications or implications of their findings for nanotechnology or agriculture.

Reply: This study provided a result of the mechanism of NPs uptake by cells and roots of A. thaliana but it is difficult to predict what will happen in field-grown plants, this will be the subject of our future studies. However, the advantages of using nanotechnology in agriculture are reported in the introduction, discussion, and conclusion sections.

10) The investigation was only carried out on one plant species (A.thaliana), which may not reflect other plant species' uptake mechanisms.

Reply: We used a model plant we expect to give insights on NPs uptake mechanisms and then conduct future analyses to generalize the mechanisms. We do not exclude extending the analysis of uptake mechanisms to other plant species.

11) The study only looked at the endocytosis mechanism of PLGA NPs, but did not investigate other potential pathways that may contribute to the uptake of NPs in plant cells.

Reply: Our previous work (Palocci et al., 2017) suggested that the preferred internalization pathways of PLGA NPs were non-clathrin pathways. In this work, we wanted to understand whether these were dynamin-dependent or independent pathways. In the future, it is our intention to conduct analyses to understand whether other pathways are involved.

12) The study relied on fluorescence imaging to visualize the uptake of NPs, which may not provide a comprehensive understanding of the mechanisms involved in NP uptake.

Reply: The approach of this work was to proceed by trying to exclude mechanisms not involved in the uptake process of PLGA nanoparticles, in this case specifically those involving dynamin.  Further analysis will be needed in the future to get closer and closer to understanding the mechanisms involved in NP uptake. Moreover, the chemical versatility of many biopolymers used for the synthesis of nanoparticles allows to improve cell targeting through conjugation with molecules capable of binding to receptors of the cell wall and membrane.

However, it is to be considered that each type of nanoparticle may have a different affinity to the cell wall and also different penetration capacity depending on composition and dimensions.  

13) The study used dynasore as an inhibitor to investigate the involvement of dynamin in the uptake of NPs, but other pathways that are not dependent on dynamin activity may also be involved in NP uptake.

Reply: The analyses provided insights into which mechanisms can be excluded from those used by plant cells and organs for PLGA NPs uptake in A. thaliana. On the basis of our results, we cannot exclude the simultaneous involvement of other dynamin-independent pathways and we will try to investigate this aspect in future studies.

14) The cytotoxicity test performed on root cells to evaluate the inhibitor cytotoxicity was limited to one dye (propidium iodide). Other methods of assessing cytotoxicity may be necessary to evaluate the safety of NP uptake in plant cells fully.

Reply: We agree with the reviewer, we also routinely use fluorescein diacetate in our experiments, but the choice of propidium comes from the work of Truernit and colleagues in which they state “the fluorescent dye propidium iodide is commonly used to stain plant roots”.  

References added to the bibliography:

  • Truernit, E., Haseloff, J. A simple way to identify non-viable cells within living plant tissue using confocal microscopy. Plant methods. 2008, 4, 1-6. 
  • Truernit, E.; Siemering, K.R.; Hodge, S.; Grbic, V.; Haseloff, J. A map of KNAT gene expression in the Arabidopsis root. Plant Mol. Biol. 2006, 60, 1.

Reviewer 2 Report

The research paper entitled "Confocal microscopy investigation of biopolymeric PLGA nanoparticle uptake in Arabidopsis thaliana L. cultured cells and plantlet roots (plants-2349499) was reviewed. After reading the manuscript, I suggest to author to revise wisely for publication in Plants. Please do the revision for this manuscript based on comments below (major revision):

1. The paper contains some grammatical errors and typo-mistakes that should be corrected. The English language should be greatly improved. For better readability of the text, the mathematical/physical symbols should be writing italics, the subscript/superscript notations should be properly rechecked, the equations should be numbered, the units should be rechecked, the illegible/non-understandable axis descriptions should be rechecked, and the abbreviations should be opened when mentioned for the first time, and so on.

2. The Abstract part should be revised. It should clearly summarize the problem, state the concept and the method, and inform the important results and conclusions in the present study. As well, it should contain some qualitative and quantitative results.

3. The introduction part should be further improved. The studied samples are based on natural nanocompoistes. These materials are very promising materials for several practical applications, which can be highlighted in the Introduction part. So, some recent references should be inserted and discussed which will be very helpful for the researchers/readers. The authors should look at the other studies and discuss them in the manuscript.

4. In the experimental synthesis part: I suggest adding a schematic flowchart or a schematic illustration for the synthesis.

5. The authors should also compare results with those done in the literature for other materials and other compounds.

6. The conclusion part should be more concise. It should summarize the important observations, major findings (in qualitative and quantitative forms), and future perspectives of the present work. Conclusions should refer NOT only to results but also to the causes of obtained results.

7. What is the conclusion of this research? What problem does this research solve in the world?

8. The experimental section needs new references.

9. The used amount of samples should be added in the synthesis section of PLGA NPs preparation.

10. The author needs to discuss the obtained result with the previously published data. This manuscript lacks discussion.

 Moderate editing of English language

Author Response

1) The paper contains some grammatical errors and typo-mistakes that should be corrected. The English language should be greatly improved. For better readability of the text, the mathematical/physical symbols should be writing italics, the subscript/superscript notations should be properly rechecked, the equations should be numbered, the units should be rechecked, the illegible/non-understandable axis descriptions should be rechecked, and the abbreviations should be opened when mentioned for the first time, and so on.

Reply: We apologize and we agree with the reviewer. The English language has been improved and the text was proofread by a native English speaker.

2) The Abstract part should be revised. It should clearly summarize the problem, state the concept and the method, and inform the important results and conclusions in the present study. As well, it should contain some qualitative and quantitative results.

Reply: As requested by the reviewer, the abstract part has been revised.

3) The introduction part should be further improved. The studied samples are based on natural nanocompoistes. These materials are very promising materials for several practical applications, which can be highlighted in the Introduction part. So, some recent references should be inserted and discussed which will be very helpful for the researchers/readers. The authors should look at the other studies and discuss them in the manuscript.

5) The authors should also compare results with those done in the literature for other materials and other compounds.

10) The author needs to discuss the obtained result with the previously published data. This manuscript lacks discussion.

Reply 3, 5, and 10: NPs of different compositions, shapes, or sizes have different physico-chemical properties thus at lines 192-221, 222-231, and 239-264 we discussed what is currently known about the cell wall size selection of NPs, uptake, and translocation in the plant. Moreover, we provided further improvements to the introduction part.

4) In the experimental synthesis part: I suggest adding a schematic flowchart or a schematic illustration for the synthesis.

Reply: Following the reviewer’s suggestion, we have added a figure with a scheme of the microfluidic reactor used for NPs synthesis.

6) The conclusion part should be more concise. It should summarize the important observations, major findings (in qualitative and quantitative forms), and future perspectives of the present work. Conclusions should refer NOT only to results but also to the causes of obtained results.

7) What is the conclusion of this research? What problem does this research solve in the world?

Reply 6 and 7:  As suggested by the reviewer the conclusions section has been improved by summarizing the achieved results.

8) The experimental section needs new references.

Reply: As suggested by the reviewer we added two more references in subsection 4.6 that justify the choice of using propidium iodide.

We decided to cite May and Leaver's work (1993) because it is used as a reference for in vitro culture of A. thaliana by several authors e.g. 

- Kruger, N. J., Huddleston, J. E., Le Lay, P., Brown, N. D., & Ratcliffe, R. G. (2007). Network flux analysis: impact of 13C-substrates on metabolism in Arabidopsis thaliana cell suspension cultures. Phytochemistry, 68(16-18), 2176-2188. For this reason, we have cited this paper.

- Pellny, T. K., Locato, V., Vivancos, P. D., Markovic, J., De Gara, L., Pallardó, F. V., & Foyer, C. H. (2009). Pyridine nucleotide cycling and control of intracellular redox state in relation to poly (ADP-ribose) polymerase activity and nuclear localization of glutathione during exponential growth of Arabidopsis cells in culture. Molecular Plant, 2(3), 442-456.

9) The used amount of samples should be added in the synthesis section of PLGA NPs preparation.

Reply: The synthesis can be performed virtually with any quantity of samples, without changing the procedure. In a typical preparation, we collected 50 mL of NPs suspension, as added, alongside concentrations and volumes, in the manuscript.